# An Improved Single Shot Multibox Detector Method Applied in Body Condition Score for Dairy Cows

**DOI:** 10.3390/ani9070470

**Published:** 2019-07-23

**Authors:** Xiaoping Huang, Zelin Hu, Xiaorun Wang, Xuanjiang Yang, Jian Zhang, Daoling Shi

**Affiliations:** 1Institute of Intelligent Machines, Chinese Academy of Sciences, Hefei 230031, China; 2University of Science and Technology of China, Hefei 230026, China; 3School of Information Science and Engineering, Ocean University of China, Qingdao 266100, China; 4School of Electronic and Communication Engineering, Anhui Xinhua University, Hefei 230088, China

**Keywords:** body condition score (BCS), sing shot multi-box detector (SSD), non-contact sensing, machine vision, dairy cow

## Abstract

**Simple Summary:**

Body condition score (BCS) is an important work for feeding management and cow welfare on the farm. The aim of our study is to assess the BCS automatically and replace the traditional manual method. In this study, we firstly built a non-contact and no-stress platform with a network camera, which can monitor the BCS of dairy cow remotely, and the back-view images of the cows were collected and the data set labeled by veterinary experts was built. Secondly, the improved Sing Shot multi-box Detector (SSD) algorithm was introduced to assess the BCS of each image. Finally, the experiments were carried out and the results showed the improved SSD had advantages of higher detecting speed and smaller model size compared with the original SSD.

**Abstract:**

Body condition scores (BCS) is an important parameter, which is in high correlation with the health status of a dairy cow, metabolic disorder and milk composition during the production period. To evaluate BCS, the traditional methods rely on veterinary experts or skilled staff to look at a cow and touch it. These methods have low efficiency especially on large-scale farms. Computer vision methods are widely used but there are some improvements to increase BCS accuracy. In this study, a low cost BCS evaluation method based on deep learning and machine vision is proposed. Firstly, the back-view images of the cows are captured by network cameras, resulting in 8972 images that constituted the sample data set. The camera is a common 2D camera, which is cheaper and easier to install compared with 3D cameras. Secondly, the key body parts such as tails, pins and rump in the images were labeled manually, the Sing Shot multi-box Detector (SSD) method was used to detect the tail and evaluate the BCS. Inspired by DenseNet and Inception-v4, a new SSD was introduced by changing the network connection method of the original SSD. Finally, the experiments show that the improved SSD method can achieve 98.46% classification accuracy and 89.63% location accuracy, and it has: (1) faster detection speed with 115 fps; (2) smaller model size with 23.1 MB compared to original SSD and YOLO-v3, these are significant advantages for reducing hardware costs.

## 1. Introduction

Body condition score (BCS) is often used as a critical measure of how effective feeding is on a farm. BCS is an important parameter, which describes the relative fatness or energy reserves of a cow, regardless of body weight and frame size [1]. By estimating BCS, it can quickly see if the cow is outside its optimal condition. Calving BCS is an important determinant of early-lactation dry matter intake, milk yield and disease incidence [2]. The sick cows in a state of abnormal BCS should be separated to make new feeding plan. BCS is also considered as an indicator of the herd productivity for farm management [3]. Gillespie et al. [4] found that feed cost accounted for 60% to 65% of the total cost, and more than 10% of the feed was wasted because of overfed. BCS contributes to healthier, more productive cows and saves on feed costs.

On the farm, BCS needs to be evaluated accurately at specific times in the lactation cycle. A view of BCS evaluation technology mainly undergoes two stages: manually method and machine automatic scoring methods. Manually method is the first stage, it is carried out by veterinary experts or skilled persons [5]. By looking at some key parts of body, such as tail, backbone, long ribs and short ribs, or even touching the fat and feeling it can easily get the BCS value. However, it takes a long time to score all the cows in a large-scaled farm, and the results may different from person to person [3,5]. Another disadvantage of this method is that it put stresses on the animals and the experts may be attacked when cows are touched or closed to. In 2012, Thorup et al. [6] proposed a semi- automated BCS system which introduced the automatic weight method when evaluated the BCS. DeMol [7] in 2013 also did the similar experiment. There exists a certain correlation between cow’s weight and its BCS, but liveweight can’t solely be used as an indicator of nutritional status of dairy cows.

With the development of computer science, BCS enter the stage of automatic scoring by machines. Machine automatic scoring methods can mainly be divided into two categories: ultrasonic measurements and computer vision methods. In literature [8,9] they both used an ultrasonic measurement device to scan special parts of body, such as ribs, pin, tail setting, and spine. Although the accuracy of BCS was extremely high, the cows should be blocked individually in self-locking manger when the experiments were carried out. This method is not suitable for larger-scale farm with thousands of cows. Actually, the ultrasonic devices can be alternated by cameras. Cow’s shape and size can be calculated from images [10,11]. In 2015, DeLaval Corporate [12] designed the first commercially available 3D BCS scanner system based on image processing technologies. It only requested the cows to pass through a narrow passage under the cameras. In 2017, Imamura et al. [13] used TOF 3D cameras to evaluate BCS. Other researchers, including Rodríguez et al. [14], Masahiro et al. [15], Hansen et al. [16], and Juan et al. [17] also used 3D cameras such as Kinect to assess BCS. Machine vision methods eliminate the guesswork and the inaccuracies of manual scoring, and the efficiency has been greatly improved. However, 3D cameras are generally expensive, and are also inconvenient for installation or position adjustment (or both).

The BCS can be treated as a target detection and classification problem in the field of image processing. With the development of deep learning technologies, BCS is actually a pattern recognition problem which can be treated as object detection and classification [18]. There are many excellent targets detection and recognition algorithms such as Faster R-CNN [19,20], YOLO [21,22], SSD [23,24] and their derived algorithms. Although Faster R-CNN and YOLO have more detecting accuracy, it cannot meet the condition of real-time application. SSD was proposed by Liu in 2017, its detecting speed is faster than the first two algorithms. Compared with YOLO, the SSD has the advantage of mean Average Precision (mAP). However, the efficiency of SSD needs to be improved, and the accuracy is high in some specific data sets, it is not yet known whether SSD can also offer such accuracy and efficiency in BCS assessing with our dataset. To answer this question, the SSD and its improved algorithm is introduced in our paper. For experimental comparison, YOLO-v3 is also introduced because it’s another more recent deep learning method based on regression. Overall, a large number of samples for model training are needed, and a common camera (with 3 megapixel, 1280 × 720 resolution, 25 fps, and working in nature light) is enough to capture 2D images of some key body parts of dairy cows [25,26].

The aim of this paper is to develop an automatic dairy cow body condition scoring system with SSD algorithms by using a cheap 2D network camera. It is a non-contact sensing method and is almost free from stress on the cows. The final output of our research is to present a reliable and suitable methods for everyday use for assessing BCS of cows. The experiments display that the improved SSD algorithm helps to increase the accuracy of BCS estimation.

## 2. Materials and Methods

The experiment is performed in compliance with China’s legislation on animal protection and approved by the Agriculture Committee of Anhui Province. The experiment location is in Huahao Ecological Farming Co., Ltd., located in Luan City, Anhui Province of China. The farm is a medium-sized modern dairy farm with 1500 Holstein cows, which provides us with sufficient sized samples to represent a commercial dairy herd.

### 2.1. Principle of BCS Assessing

BCS can be done using only visual indicators or a combination of visual and tactile estimation of key bone structures for fat cover. Tactile estimation may be a little difficult, it can be done during routine processing of cows through a corridor. The key areas or body parts for BCS assessing are the backbone, pins, tailhead, long ribs, short ribs, hips, and rump, as shown in Figure 1. Evaluating cows for fatness through the ribs, tailhead and the backbone will help refine veterinary expert’s skill to visually access the body condition.

There are many different scale scores in the world. In the USA, a 5-point scale system proposed by Windman [27] was commonly used, where BCS value varies from 1 to 5. In Australia, Earle [28] proposed an 8-point scale, which was widely used for dairy cattle, with a score of 1 being extremely thin and 8 being very obese. Macdonld et al. [29] proposed a 10-point scale system in New Zealand. In this study, we adopted 5-point scale system which was widely used in the world. In order to achieve higher BCS accuracy, the intermediate. 5 point was employed.

In 5-point scale system, BCS 1 means the cow is very thin and physically weak, cows in the status may be unhealthy. BCS 5 represents the cow is in over-weight status and it increases the costs because of overfed. BCS 3 is the ‘ideal’ condition that cows are in a good overall appearance. 5-point scale is very easy to master. However, for the subjectivity error, observers judge the same cow can lead to different scores. BCS assessing by machine vision method is very necessary to eliminate these manual mistakes.

### 2.2. Data Collection Platform

Figure 2 shows the developed system on the farm of Huahao Ecological Farming Co., Ltd. In the system, the network camera is needed. The model of the camera is DS-2CD3T56DWD-I5, with 5 million pixels, which is made by Hikvision Company (Hangzhou, China). It worked in an indoor environment with natural light, and it aimed downward to the milking passage. The frame frequency was set to 25 FPS (Frame Per Second), and the resolution and size were set to 1080 × 720 dpi. It is a common 2D camera which is relatively inexpensive compared with other 3D cameras, such as Kinect. Another advantage of the camera is that the equipment with RJ45 interface can directly communicate with Ethernet server which can monitor the cows remotely without any stress on the animals. The camera was installed on a steel beam which was 2.6 m high from the ground. Under the camera, there was a narrow milking passage allowing the cows to pass in line.

The distance between the cow’s back and the camera was about 1.4 m. When these Holstein cows walked voluntarily below the Hikvision camera, the back view of the cow images (or videos) were captured. Therefore, the key areas of cow body can be clearly displayed in the picture. In the power room of the farm, we placed one hard disk video recorder of which the storage capacity is 4 TB. Hence, the image data can be got either from internet or copying from the hard disk. We captured three times every two weeks, during the acquisition of the cow images, two experts were standing aside and rating the BCS of the cows simultaneously.

The farm has a strict management system, and outsiders cannot enter the core area of milk production, such as milking passage in Figure 2. On this occasion, the data can be transmitted to our laboratory via the internet. We can remotely monitor the cows when they walk through the milking passage every day. The non-contact and non-stress platform for assessing BCS of dairy cows had been built, and the next stage was to process the images and carry out the deep learning algorithms.

### 2.3. Image Samples and Dataset

In order to train and validate the deep neural networks, the samples and a dataset should be built. Tens of thousands of images were captured by the remote monitoring platform, and 8972 images containing cow tail were selected for further processing. Firstly, these images should be mapped to the BCS recorded by the experts. Then all the 8972 image samples were labeled manually by the tool of LabelMe, which was an open annotation tool to build image databases for computer vision research, developed by computer science and artificial intelligence laboratory (CSAIL) of Massachusetts Institute of Technology (MIT). The information of each labeled image was saved on an annotated XML file which followed the Pascal VOC data format.

We select three distinct images, which are the back-view images of dairy cows, as shown in Figure 3, each representing a specific body shape. In the figure, the key area is the tail, but it should be mentioned that the tail is not only the tail itself, it covers most area of caudal body of dairy cow, such as tailhead, pins, and some backbone. This area is highlighted with a green rectangle. There is a close linear correlation between the BCS and the area of rectangle (it is also called ground-true box).

These images were split into training and test sets. In this sense, 90% of the images (8074) were used for model development (for training) and 10% of the images (898) were used for model validation (test). The two datasets were composed of BCS values ranging from 2.5 to 4.5 preserving samples distribution of the whole dataset. The detail of the BCS distribution for each BCS level is shown in Table 1, and the proportion of BCS values are shown in Figure 4. The last row of the table is the total value of each item. Finally, this dataset is named as Cow Tail Dataset (CTD for short), which will be used frequently in the following sections.

It should be noted that, all cows on the Huahao farm were well fed and in healthy status. There were no milking cows with BCS value below 2.5. The cow’s BCS below 2.0 is considered as sick or unhealthy cow. Similarly, the cows with BCS 5.0 are overfed.

### 2.4. BCS Estimation Models

#### 2.4.1. Original SSD Model for BCS Assessing

The SSD is a relatively fast and robust method. It is based on a feed forward convolution network that makes full use of the features of different output layers for object detection. The network structure used for BCS assessing for dairy cow is shown in Figure 5. It can be divided into two parts: the VGG-16 and the extra feature layers. The front of the network for BCS classification is VGG-16, which is the baseline network with 16 layers including 13 convolution layers and 3 Fully-Connected (FC) layers. The filters of all layers are used with a very small receptive field: 3 × 3, which is a main contribution to improve the classification ability and decrease the amounts of parameters. The second part of the network is extra feature layers. There are six different scales of feature maps to detect different size of objects. Low-level layers such as FC6 and FC7 are used to detect small targets, and high-level layers such as Conv10_2 and Conv11_2 are used to detect targets of large size.

In the structure of SSD, when entering a back-view image of the cow and its label file, it uses a feature extraction network to generate a different size feature graph in which a 3 × 3 filter is used to evaluate the default box. After producing the default box, it will predict the migration and the probability of classification (BCS). Each feature map cell predicts the object BCS and migration of several default boxes, for the *k* default boxes at some given location. And then the *c* classification scores and 4 position offsets related to the ground-true box are calculated. The extra feature layers, in Figure 5, are often added to the end of VGG-16. These layers help to decrease in size progressively and allow detecting predictions at multiple scales.

The detection results of different layers are merged in detection block which can decides if the area exists the targets and assigns predicted bounding boxes. While the non-maximum suppression (NMS) block is used to suppress redundant detection boxes. The objects are detected directly in the feature maps of each layers, consequently, there is no need to generate the candidate regions. Thus, that is why SSD can run at a high frame rate.

#### 2.4.2. The Improved SSD Model

The original SSD performs very well in terms of both speed and accuracy by fully utilizing the feature maps of different layers. However, each layer contains its unique feature information, and is used independently, the close relationships between different layers is not well considered. On the other hand, the default boxes do not correspond to the actual receptive fields of each layer, and a better size of default boxes is effective so that its position and scale are better aligned with the receptive field of each position on a feature map [24]. For these concerns, firstly, we introduce the DenseNet [30] to improve the thin connection of original SSD between different layers, it can help to reduce information loss. Secondly, inception-v4 [31,32] block is employed to expand receptive field and reduce parameters of networks. The improved network structure used for BCS assessing is shown in Figure 6.

(1) Replacing VGG-16 with DenseNet

In order to improve the performance of original SSD, in our study, we introduce the DenseNet that means the basic network VGG-16 is replaced by DenseNet. DenseNet is well considered the connection between any two layers with the same feature-map size. Instead of drawing representational power from extremely deep or wide structure, DenseNet exploit the potential of network through feature reemployment, producing condensed models, which are easy to train and highly parameter-efficient. Connecting feature maps that are learned by different layers enhances variation in the input of subsequent layers and improves efficiency.

The DenseNet [30] consists of two parts: dense blocks and transitions. There are 3–6 dense blocks (we fix 6 dense blocks in our method) in the improved SSD networks. In the dense block as shown in Figure 6, it comprises *L* layers, each of which implements a non-linear transformation $H_{\ell }()\;$, where ℓ represents the layer. The input of each layer come from the outputs of all the preceding layers. Thus, it produces *L*(*L* + 1)/2 connection in each block. The advantage of this dense connection ensures that the information flowing throughout each layer is more complete. Considering one image $x_{0}$ passed through a convolutional network, we assume that the $\ell^{th}$ layer receives the feature maps of all the preceding layers $x_{0}$, …, $x_{\ell -1}$, as shown in the following formula:(1)$$x_{\ell }=H_{\ell }\left(\left[x_{0},x_{1},\ldots ,x_{\ell -1}\right]\right)$$
where $\left[x_{0},x_{1},\ldots ,x_{\ell -1}\right]$ represents the connection of each feature map produced in layer 0, …, ℓ−1. $H_{\ell }()\;$ is a composite function of three consecutive operations: batch normalization (BN), a rectified linear unit (ReLU), and a 3 × 3 convolution [31]. Therefore, in our method, the relationships between each layer of the feature maps are well considered. The combination of rich details information in the low-level layers and strong semantic features in the high-level layers helps the new SSD method to fuse more features.

The second important part is transition which consists of a 1 × 1 Convolutional layer and a 2 × 2 average Pooling layer, as shown in Figure 6, transition is the layer between dense blocks. For the connection operation (Equation 1) is not capable when the size of feature maps has been changed, convolutional networks are down-sampling layers that help to resize the shape of each layer. Thus, the shape of feature maps always keeps 2H × 2W that is equal to that of the target layer. The 1 × 1 convolutional layer is used to unify the channel dimension to 512. The function of average pooling layer is to reduce the amount of the parameters and computation in the network. At the end of the last dense block, a global average pooling is employed and a target detector is attached. The non- maximum suppression block is used to remove the redundant boxes in the detection task. The final output of the network is the BCS level of the given images.

(2) Introducing Inception-v4

There are 12 convolution layers in each dense block. In our study, we replace these common convolution layers with 6 Inception-v4 [32] blocks, the schema of each block is shown in Figure 7. Although VGG-16 in SSD method has the compelling feature of architectural simplicity, it comes at a high cost for a lot of computation. However, the computational cost of Inception-v4 is much lower than VGG. The parameters employed by Inception-v4 are also less than VGG. These factors meet the needs of high real-time requirements where memory or computational capacity is inherently limited. In order to design a compact, low-cost BCS assessing system, inception-v4 block is introduced to replace the single 3 × 3 convolution kernel. The overall schema of the Inception-v4 network in each dense block is shown in the Figure 7.

The network design of Inception-v4 follows the previous version (Inception-v2/v3 [32]). The filter concatenation functions as connecting the images according to depth. For example, the output of three 10 × 10 × 3 images is 10 × 10 × 9 by filter concatenation. The training speed of this module is greatly accelerated by combining with variants of these blocks (with various number of filters). Meanwhile, The Inception block expands receptive field and reduces parameters of the dense blocks. To see the effect of these improvements, the contrast experiments are carried out in Section 3.

(3) The Architecture of Our SSD

The proposed SSD algorithm is a muti-scale proposal free detection framework which is similar to original SSD. And the framework of our SSD is built on the original SSD, therefore, the speed and accuracy advantages will be inherited obviously. In our method, the advantages of DenseNet and Inception-v4 are absorbed, the new network architecture is formed in Table 2. There are 6 dense blocks and 6 transition layers, and each dense block contains 6 inception blocks (as shown in Figure 7). The output size and the connection of each layer detail in the table.

### 2.5. Performance Evaluation

BCS assessing is essentially a problem of target detection and classification. A group of metrics are used to evaluate the performance of models. The following seven important terms that need to be introduced, they are also commonly used in other literature [16,17].

Intersection Over Union (IoU): It evaluates the overlap between two bounding boxes. It requires a ground truth bounding box $B_{gt}$ (as shown in Figure 3, our labeled rectangle) and a predicted bounding box $B_{p}$ (that is our algorithms predicted rectangle).
(2)$$IoU=\frac{area\left(B_{gt}\cap^{\text{​}}B_{p}\right)}{area\left(B_{gt}\cup^{\text{​}}B_{p}\right)}$$Mean IoU: It is the mean value of IoU for all the test sample images.
(3)$$mean\;IoU=\frac{1}{N}\sum_{i=1}^{N}\frac{area\left(B_{gt}^{i}\cap^{\text{​}}B_{p}^{i}\right)}{area\left(B_{gt}^{i}\cup^{\text{​}}B_{p}^{i}\right)}$$Classification Accuracy (CA): it represents the effectiveness of a classifier. Its calculating method is given in Equation (4) where TP is the abbreviation of true positives, TN is true negatives, FP is false positives, and FN is false negatives.
(4)$$CA=\frac{TP+TN}{TP+FP+TN+FN}$$Mean Average Precision (mAP): it is a metric to measure the mean accuracy of N classes (N is the class number for all), which is different from AP (for one class).Confusion Matrix: in order to assess the accuracy of an image classification, creating a confusion matrix is common practice. It identifies the nature of the classification errors, as well as their quantities.Frame Per Second (FPS): it is a speed measure for the network model to run each image frame. Second per frame (SPF) is also another form of speed metric to evaluate the performance of the models.Model Size: it is a storage space measure (in MB) for the validated model after training work. This parameter has a significant importance when the network model is transplanted into the embedded system. The smaller the model is, the lower the system costs.

## 3. Training

In order to verify the validity of the proposed SSD target detection algorithm, we carried out three algorithms that are our SSD, the original SSD, and YOLO-v3. The training data set in Table 1 was used to obtain network models, and the test data set was used to validate the performances of these methods.

For an m × n feature map, (c + 4) × k × m × n outputs will be generated. Assuming $x_{ij}^{p}$ = 1 represent for the matching of *i*-th default box and the *j*-th ground-true box. If $x_{ij}^{p}$ = 0 means the there is no matched boxes. According to this rule, if $x_{ij}^{p}$ ≥ 1, it means that there are more than 1 default boxes match the *j*-th true label. The loss function (L) of the SSD used for BCS assessing is defined as follows:(5)$$L\left(x,c,l,g\right)=\frac{1}{N}\left(L_{conf}\left(x,c\right)+\alpha L_{loc}\left(x,l,g\right)\right)$$
(6)$$L_{conf}\left(x,c\right)=-\sum_{i\in Pos}^{N}x_{i,j}^{p}\log\left({\hat{c}}_{i}^{p}\right)-\sum_{i\in Neg}\log\left({\hat{c}}_{i}^{0}\right)$$
(7)$${\hat{c}}_{i}^{p}=\frac{\exp\left(c_{i}^{p}\right)}{\sum_{p}\exp\left(c_{i}^{p}\right)}$$
where L is number that the default box and the ground-true box has been matched. α is the weight parameter which is generally set to 1 by cross validation. c is the confidence level of softmax function for each class, l is the parameters (including the center position coordinates, the with and the height) of default box, g is the parameters of ground-true box. p is the class number. $L_{conf}$ is classification confidence loss over multiple classes confidences (*c*). $L_{loc}$ is the loss of the position regression, containing the center position of the bounding boxes as well as the width and height.

A lot of detection boxes were produced when using the multi-scale feature layers to detect key parts of cow body, while most of them had no target. So, we set a confidence threshold as 0.6 to eliminate detection boxes with low confidence. Then the extra boxes were eliminated by non-maximum suppression (NMS), thus the optimal location of the cow body was found.

The Caffe framework were used to train the deep learning model. Training was carried out on a 64 bits Ubuntu 16.04 computer with Intel Xeon E5-2687W CUP @ 3.4 GHz × 4 and NVIDIA GTX1080Ti GPU, and 32 GB RAM, and the total time for training was two days for each algorithm. In the next step, we will analysis the location accuracy, classification accuracy and the running-time (speed by FPS) for one single image for comparison. This work will be carried out in Section 4.

## 4. Results and Analysis

### 4.1. Position Results and Comparison

After the training work, the model of networks was validated. To see the location performance, nine image samples are randomly selected from the test dataset to compare the performance of the three algorithms. The ground-truth bounding box is drawn in red, our SSD predicted bounding box is drawn in yellow, the original SSD predicted bounding box is drawn in purple, and the YOLO-v3 predicted bounding box is drawn in green, as shown in Figure 8. The score assessed by each network model is also marked next to the box. It is clear that all the predicted bounding boxes by the three methods are generally closer to the ground-truth box.

For a more detailed comparison, the accuracy and other metrics need to be taken into consideration. IoU is a parameter that evaluates the overlap rate between the ground-truth bounding boxes and the predicted boxes by the deep learning methods. In order to get the IoU comparison, all the image samples in the test set are calculated, and 50 images are selected to present the IoU curves in Figure 9. By integrate analysis of curves, all the three curves are above 60%, and the IoU of our SSD in red and the original SSD in green seem to be higher than that of YOLO-v3 method in blue. For the 1795 samples in the test set of CDTs, the means of IoU are taken and the results are provided in Table 3. From the table, a conclusion can be drawn that the mean IoU of Our SSD with 89.63% and original SSD with 90.55% are higher than 83.64% of YOLO-v3 method. However, the mean IoU of our SSD is 0.92 percentage point lower than the original SSD.

### 4.2. BCS Assessing Results and Comparison

In order to verify this performance of the above models, classification accuracy (CA) is employed. Table 4 shows the CA for each class (or BCS values) in the test set. For each BCS level, a conclusion can be drawn that the extreme BCS levels (referred to BCS 2.5 and BCS 4.5) have a higher accuracy than the middle BCS levels in all the three algorithms. The last row of Table 4 is mAP for each method. The mAP of our SSD is 98.46%, the mAP of the original SSD is 99.10%, both of them have a higher accuracy than YOLO-v3 of which the mAP is 88.84%. The mAP of our SSD is a little (0.64%) lower than the original SSD.

In order to assess the accuracy of an image BCS classification, it is also common practice to create a confusion matrix. Table 4 shows the confusion matrices of test samples classification for BCS assessing. In confusion matrices, the BCS assessing results by the algorithms are compared to additional ground-truth information. The ground-truth BCS values can be found in the rows of the confusion matrix, and the predict results by algorithms are appeared in the columns. The main diagonal of the confusion matrix represents the classification results by the algorithm exactly fit the ground-truth BCS values. Particularly, green cells represent exact predictions, orange cell represent predictions with 0.5 units of error, and red cells represent predictions with 1.0 units of error.

From Table 5, it can be perceived that there are only a few errors with 0.5 units predicted by our SSD and original SSD. But the error is serious in YOLO-v3, it even exists the errors with 1.0 units in red cell in the table. For example, in first column (BCS = 2.5), there are 99 images are accurately recognized as 2.5 and one image wrongly sorted into 3.0 category by our SSD algorithm. The same result can be got by original SSD. In YOLO-v3 cell, one image of BCS 2.5 is wrongly sorted into 3.5 category with 1.0 units of error. All the other columns are similar with the first column.

### 4.3. Running Speed and Model Size Analysis

Deep learning methods always cost much computing time, for this disadvantage, the real-time application requires hardware with high configuration. One effective way is to reduce the computing time. In our study, the dense blocks and inception-v4 block are employed to improve the original SSD method. The first row of Table 6 shows the running speed of our improved SSD reaches to 115 FPS，which is more faster than original SSD (with 39 FPS) and YOLO-v3 (with only 17 FPS). It is very important for the practical application, especially for low-end application with less CPU performance. For another important improvement is the model size which measures the memory size of the trained networks. The second row of Table 6 show our SSD takes the minimal storage space with just 23.1 MB, reduced by more than 10 times compared with YOLO-v3 (246 MB). The proposed SSD also has obvious improvement compared with original SSD with 97.1 MB. For these advantages of our method, it is very practical and requires low on hardware, requiring less on CPU and memory. Thus, it may turn this BCS system to be cheap.

## 5. Discussion

### 5.1. Why the Detection Totals are Different from the Test Data Set?

There is an interesting phenomenon between Table 1 and Table 5. In Table 1, the totals per BCS (score) are 99, 254, 212, 207 and 126, but the sum of each BCS level in confusion matrices in Table 5 are not the same, e.g., they are 100,255, 215, 210 and 127 in our SSD cell, and they are 99,253, 212, 208 and 126 in the original SSD cell. In deep learning algorithms, there are predict bound boxes for each image with a certain threshold, which is set to 0.6 in our study. Under this condition *threshold* ≥ 0.6, three situations may happen: (a) predict bound box number N > 1, there are more than one predict bound boxes for this image. It results in that the totals per BCS in Table 5 are more than the totals per BCS in Table 1; (b) predict bound box number N = 1, there is only one predict bound box for this image, if all the images in test set are like this, the totals per BCS in Table 5 are equal with the totals per BCS in Table 1; (c) predict bound box number N < 1, there is no predict bound box for this image. This situation caused that the totals per BCS in Table 5 are less than the totals per BCS in Table 1.

### 5.2. Can Veterinarians Score on Images?

BCS can be rated by digital images [33,34]. The image data set labeled by veterinarians is very important for deep learning methods. Until now, there are some research achievements in assessing BCS by machine learning [14,15,16,17], they need the help of veterinarians or experts to score the cows, human empirical rules are the basis for machine learning. In reference [14] and [17], when the image data was acquiring, an expert was arranged to score the BCS of cows in situ. In our study, we made the same arrangement, two veterinary experts were asked to stand beside the milking corridor, and to record the BCS value of each cow. After this work, the video can help them to revise the scores moreover. The key parts in Figure 3 were also labeled manually by the two experts. Even we did these arrangements, the experiment results show an interesting phenomenon that the error rates are higher for BCS from 3.0 to 4.0 in Table 4 and Table 5, compared with extreme BCS of 2.5 and 4.5.

Ferguson et al. [33] found that scorer with less experience was within 0.25 units of the modal BCS 65% of the time and within 0.50 units 84% of the time. Bewley et al. [34] found that BCS changes of 0.25 cannot realistically be detected, even with trained experts. So, in our experiment, the extreme BCS levels corresponding to the cows are extremely thin or fat, so the features of the body shapes, the tail and other key parts of the cow’s body are easier to learn. The middle BCS level (referring the BCS from 3.0 to 4.0), there may be some manual error when veterinary experts rating and labelling the dataset. This subjective scoring error is inevitable, but if more veterinary experts joint the scoring work, the error may be lower in the future.

So, can veterinarians score on images? We believe that more than one veterinarian rate BCS on the scene, and aid with video (or images) can greatly minimize subjective errors. 

### 5.3. More Training Data vs. More Accurate Data

A traditional idea in the machine vision is that object detection and classification task may be solved with deeper and larger neural networks with massive training data like VOC, COCO, and ImageNet [35]. So, more and more large-scale datasets have been released in the past few years, like Open Image dataset [36], which is 7.5 times larger in image number and 6 times larger in categories than that of ImageNet. Similarly, if our cow dataset is expanded, we definitely agree that, the BCS classification accuracy will be perform extremely well. Currently, there are 8972 cow images with labeled tail, and the number will grow with time going on. 

For another, unlike other large image data set, there is no universal dataset of cow images that allows for a standardization of experimental factors. Two veterinary experts are asked to help us scoring the body condition of cows. There may be some manual error when veterinary experts labeled the dataset, especially for the middle BCS level (referring the BCS from 3.0 to 4.0). However, the extreme BCS levels (referred to BCS 2.5 and BCS 4.5) have a higher accuracy. This is why we need to improve dataset quality. If more veterinary experts joint the scoring work, the error may be lower in the future.

As a result, we both need to enhance the dataset quality and increase training images to improve the BCS assessing accuracy.

### 5.4. Future Work

The output of the experiment will be a remote monitoring system which can assess the BCS for dairy cows every day. It is a cheap system only with one common network camera and a mini industrial computer which can run the neural network model. In the near feature, we need to carry out the following three tasks to improve the system in the future work.

For lacking of unhealthy or over-fed dairy cows, he BCS below 2.5 and BCS 5 are lacking, a complete 5-point scale BCS system need to be consummated.The performance of the speed and accuracy still need to be further improved, especially for BCS from 3.0 to 4.0. The shape of cows in this section are easily confused when experts scoring the image samples. If more veterinary experts joint the scoring work to eliminate subjective error, the CA of each BCS level may be increased. Although the CA of our SSD is 98.46% that it can be improved a little more. An effective way to improve it is that the monitoring system should be lasted over a long period of time, and the image samples size should be expanded.Because of the unresolved identification, the BCS of the individual cow cannot be picked out from the database. In order to connect BCS score to the ID of each cow, the combination of camera and wearable sensors, such as pedometer and electronic ear tag may be a good solution in future.

## 6. Conclusions

The BCS of dairy cows is important for farm management. In this study, we introduce an automatic BCS assessing method with computer vision technology, and then we carry out the experiments on a low cost BCS platform with a common IP camera. The experimental results show that the accuracy of our improved SSD algorithm can reached 89.63% in cow tail extraction, and the accuracy of BCS assessment is 98.46% on average. Although the accuracies are a little lower than the original SSD, our improved SSD has more fast detection speed with 115 FPS and smaller model size with 23.1MB. The two better performances prove that our improved SSD, with the dense connection block and the Inception-v4 block, is effective. This improvement expands the reception field of SSD and reduces the amount of calculation, consequently, the running speed can be increased and model size can be smaller. These advantages are very practical when considering the hardware costs: our SSD is a better choice compared with original SSD and YOLO-v3. However, there are some disadvantages in our solution. e.g., BCS 2.0 and below are lacking, and the BCS should be connected to the ID of the individual cow. We will continue to improve it in future work.

## Figures and Tables

**Figure 1 animals-09-00470-f001:**
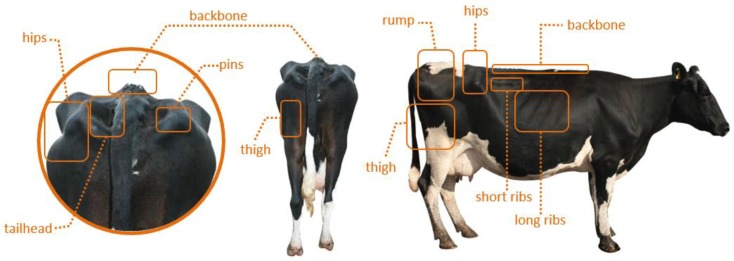
Key areas of cow body for Body condition scores (BCS) assessing. The left half shows the back view of the cow. The right half shows the side view of cow.

**Figure 2 animals-09-00470-f002:**
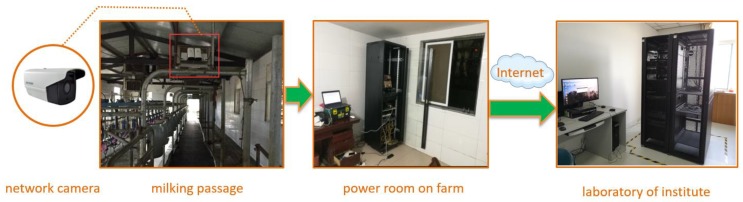
Network camera and equipment for remote monitoring platform.

**Figure 3 animals-09-00470-f003:**
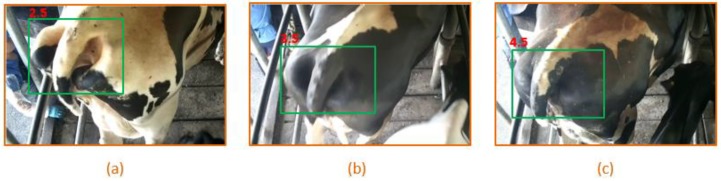
Image samples of cows back views, varying from different BCS values. The green boxes in the figures are labeled manually, we call the green box as ground-true box. (**a**) The value of BCS is 2.5; (**b**) The value of BCS is 3.5; (**c**) The value of BCS is 4.5.

**Figure 4 animals-09-00470-f004:**
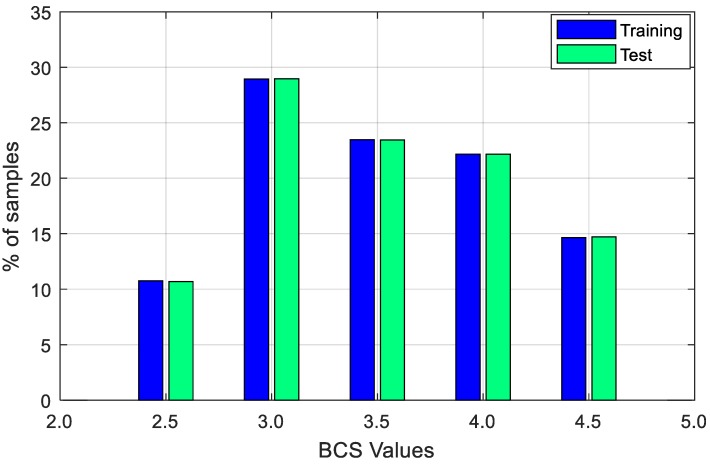
Percentage of BCS values distribution over training and test set in Cow Tail Dataset (CTD).

**Figure 5 animals-09-00470-f005:**
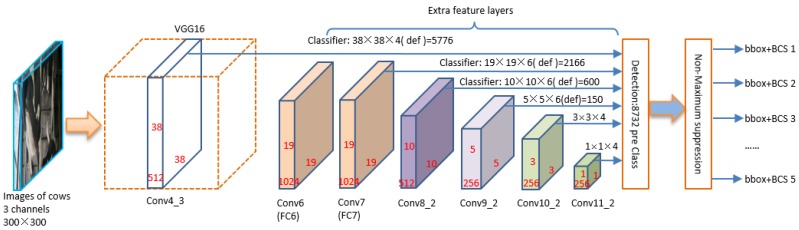
The structure of original Sing Shot multi-box Detector (SSD) method used for BCS assessing.

**Figure 6 animals-09-00470-f006:**
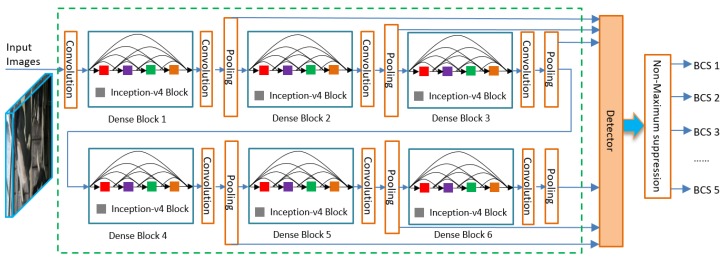
The improved SSD model used for BCS assessing.

**Figure 7 animals-09-00470-f007:**
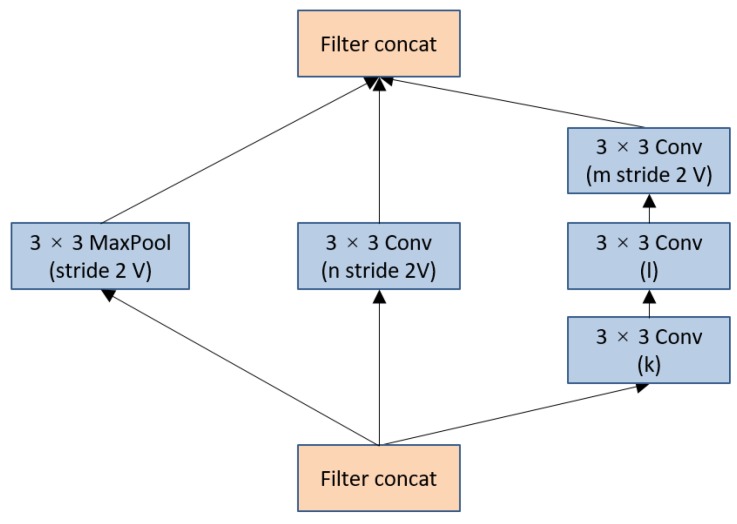
The schema of inception-v4 block for 35 × 35 to 17 × 17 reduction module. It was first referred in each dense block in Figure 6. The k, l, m, n numbers stand for filter bank sizes, with the values corresponding to 192, 224, 256, 384. (comcat is short for concatenation, and conv is short for convolution).

**Figure 8 animals-09-00470-f008:**
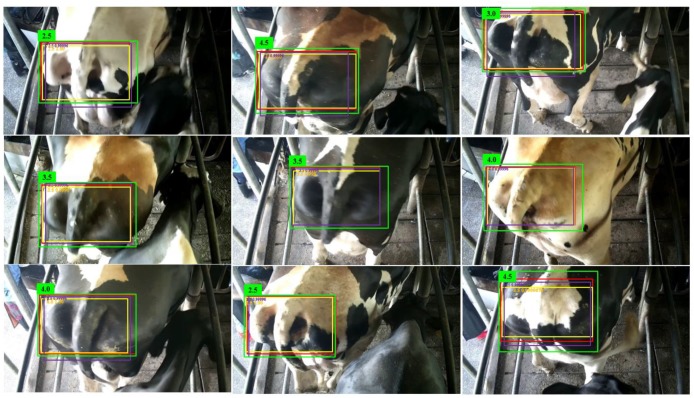
Examples of cow tail detection on the CTD test data set using our SSD, the original SSD and YOLO-v3 method. Each output box is associated with a category label and a BCS score in [2.5, 3.0, 3.5, 4.0, 4.5], one color corresponds to one kind of method in that image.

**Figure 9 animals-09-00470-f009:**
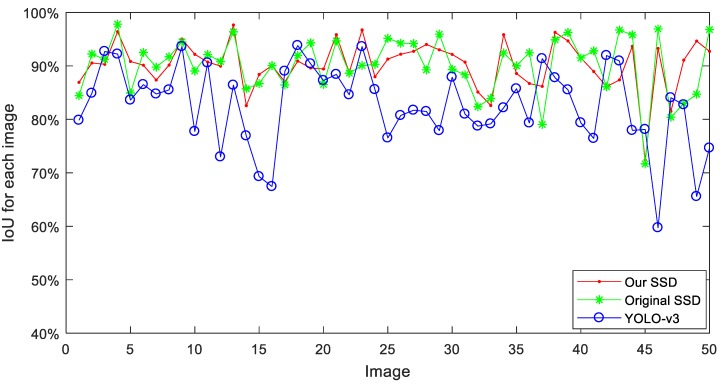
Comparison of Intersection Over Union (IoU) for each test image between our SSD, original SSD and YOLO-v3 method. To display the detail value points, only 50 images sample are randomly selected in test set.

**Table 1 animals-09-00470-t001:** Body condition scores (BCS) values distribution for training and test images.

BCS Value	Training Images	Test Images	Total Images
2.5	865	99	964
3.0	2343	254	2597
3.5	1894	212	2106
4.0	1782	207	1989
4.5	1190	126	1316
Total	8074	898	8972

**Table 2 animals-09-00470-t002:** Our improved Sing Shot multi-box Detector (SSD) architecture.

Layers	Structure	Output Shape
Input	448 × 448	448 × 448
Dense Block (1)	Inception block × 6	224 × 224 × 32
Transition Layer (1)	1 × 1 conv; 2 × 2 max pool, stride 1	
Dense Block (2)	Inception block × 6	112 × 112 × 64
Transition Layer (2)	1 × 1 conv; 2 × 2 max pool, stride 1	
Dense Block (3)	Inception block × 6	56 × 56 × 128
Transition Layer (3)	1 × 1 conv; 2 × 2 max pool, stride 1	
Dense Block (4)	Inception block × 6	28 × 28 × 128
Transition Layer (4)	1 × 1 conv; 2 × 2 max pool, stride 1	
Dense Block (5)	Inception block × 6	14 × 14 × 128
Transition Layer (5)	1 × 1 conv; 2 × 2 max pool, stride 1	
Dense Block (6)	Inception block × 6	7 × 7 × 128
Transition Layer (6)	1 × 1 conv; 2 × 2 max pool, stride 1	
Prediction Layers	Plain/Dense	–

**Table 3 animals-09-00470-t003:** Comparison of the mean Intersection Over Union (IoU) for the three methods.

Our SSD	Original SSD	YOLO-v3
89.63%	90.55%	83.64%

**Table 4 animals-09-00470-t004:** Results of the classification accuracy (CA) for each BCS level.

BCS	YOLO-v3	Original SSD	Our SSD
2.5	98.99%	98.99%	99.0%
3.0	90.49%	99.60%	98.83%
3.5	87.75%	98.58%	98.14%
4.0	79.37%	98.56%	97.67%
4.5	99.10%	100%	99.21%
mAP	88.84%	99.10%	98.46%

**Table 5 animals-09-00470-t005:** Confusion matrices of test samples classification for three algorithms. Green cells represent exact predictions, orange cell represent predictions with 0.5 units of error, and red cells represent predictions with 1.0 units of error.

Algorithm	BCS	2.5	3.0	3.5	4.0	4.5
Our SSD	2.5	99	0	0	0	0
	3.0	1	254	0	0	0
	3.5	0	1	211	5	0
	4.0	0	2	4	205	1
	4.5	0	0	0	0	126
Original SSD	2.5	98	1	0	0	0
	3.0	1	252	1	0	0
	3.5	0	0	209	3	0
	4.0	0	0	2	205	0
	4.5	0	0	0	0	126
YOLO-v3	2.5	98	3	0	0	0
	3.0	0	257	11	1	0
	3.5	1	24	179	30	0
	4.0	0	0	12	200	1
	4.5	0	0	2	21	110

**Table 6 animals-09-00470-t006:** Comparison of running speed and model size.

Performance	Our SSD	Original SSD	YOLO-v3
Running speed	115 FPS	39 FPS	17 FPS
Model size	23.1 MB	97.1 MB	246 MB

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
