# Peer review of "An Improved Single Shot Multibox Detector Method Applied in Body Condition Score for Dairy Cows"

_animals, 2019, doi:10.3390/ani9070470_

Round 1

Reviewer 1 Report

The paper describes a very interesting and important topic of BCS measuring. In general the paper has a specific focus on the used AI technology and misses sometimes some practical insights to develop a BCS system that can be used in farms. An example e.g. that is not covered is how the final BCS score is connected to the ID of the individual cow. Sometimes it is difficult to understand the specific message of sentences. Therefore I advice also to correct it for English. The following comments are restricted to the research method and the interpretation.

L2:  Why not use full text of SSD  'Single Shot Detector'

L19:  do not use the word production. The system monitors Body Condition

L26: are veterinarians the only people scoring in the traditional methods or can it also be done by other  people like farmers.

L28: change 'wildly'  in 'widely'

L57: do you have some reference to the 'person to person' differences. This is very important for your research, since you worked with two veterinarians, but no information is given on their differences and quality to score BCS. Just being a veterinarian might not be enough qualification.

L88: can you elaborate on the 'experimental comparison' since this is the only explanation why you need more algorithms. 

L90: is the statement 'a regular camera is enough' enough or do you have some minimum requirements on your images and that 2D camera with enough fps, resolution and light is in line with your requirements.

L95-106: This text is not needed in the introduction.

L115-116: The first sentence is  a repletion and not needed in this context.

L123: change (a) and (b) in left and right. You can argues whether you need the (b) part since the labelling is done based on the (a) part.

L128-129: you adopt the 5 point American scale because it only has 5 points! Is this reason enough. Based on this choice I have a more general point. You adopt the 5 point scale, but in your results you also present the intermediate .5 points (see figure 4 etc). This is not in line with each other.

L142-149: how were the practical situations where the camera is used. Was there enough light, a good FOV and how many frames per second were measured. This might also be a good place to say something on the connection between the camera measurement and the ID of the cow.  Is it correct that you have the chance with 1500 cows and three times of milking per day to extract 4500 observations per day. You work with 8972 images and measured a couple of months, which means that most of the images could not be used. Please elaborate on this aspect in material and method or in the discussion.

L168: 'we select 3 distinct images'  who is doing this and how this is done. What were the criteria to select the images.

L172: The angle of the camera and the observations seem to deliver pictures from an angle that observers are not used to score BCS. How did the observers cope with this and were they capable of scoring BCS in a proper way.

L177: Labelling was done manually. This means that the two veterinarians only looked at the images and that there is no real BCS score in the barn itself. You have to mention something on this issue. If you later on talk about specificity etc then it should be clear that it is compared to the observation of human looking at images and not compared to humans doing the BCS measurement in the barn. 

L207: Is the top line on 'extra feature layers'  too short and should it cover also the blocks Cov6 an Conv7

L223:  the whole paragraph on the improved SSD model is a very technical description of the SSD model and for animal scientist difficult to read and to judge whether it is smart to do it in this way.

L297: This paragraph  gives too much detail. Sometimes a good reference should be enough. Looking to the results more performance indicators are described in this paragraph. Omit the ones that you are not using in the results. 

L335:  why these 4 algorithms and give a good reference to the 3 existing. 

L359-360:  this seems to be part of Material and Method

L383:  as with other figures. Give also description of the axis. What are the units? you present.  Why only the results of three algorithms. What happened with the mobile-SSD algorithm.

L385: Why use a pie chart. The same information can be presented in the text or in a short table.

L390-399: this is a mix between results, discussion and  even conclusion. Stick to present the results in this part of the paper. Do the discussion in chapter 5.

L413:  The table might be misleading since you give absolute numbers. Then I expect that the totals per BCS (score) are 99, 254, 212, 207 and 126. There are different totals between the models. Think of presenting it as percentages and explain well in the text why you have different total.

L421: I cannot see in the table where the BCS 2.5 is wrongly sorted into 3.5 for the original model. Maybe you are talking about the YOLO-v3 model.

L442: what are the units in the axis.

L446-450:  is this summary needed?

L452-466:  this looks more a explanation of the results and is hardly a discussion. This seems to be for the whole discussion chapter. Try to bring in some references and e.g. can veterinarians really score on images.

L501: 'manually scoring stage by veterinary experts'  say that they score images and not in the barn.

L511-513: this sentence is the conclusion. The rest of this paragraph reads more like a summary. So it can be much shorter. 

L533: In the references 7, 12, 13,22, 24, 26, the words et al are used. Why not give all the names of the authors.

L561: It seems that a new reference starts at Thi Thi Zin.

L566: is it sufficient with website reference (see also 29 and 30) to give the link, or is it also needed to give the extraction dat.

Author Response

Thank you for your review and for your comments concerning our manuscript entitled “An improved SSD method applied in body condition score for dairy cows”. Manuscript id is animals-527248. Those comments are all valuable and very helpful for revising and improving our paper, as well as the important guiding significance to our researches. We have studied comments carefully and have made correction which we hope meet with approval. The main corrections in the paper and the responds to the reviewer comments were highlighted in red in the new manuscript. We will keep in touch with you and submitt other needed materials.

The main corrections in the paper and the responds to the reviewer comments are as following: (The more detailed responses with pictures are in the attachment.)

(1)  The paper describes a very interesting and important topic of BCS measuring. In general the paper has a specific focus on the used AI technology and misses sometimes some practical insights to develop a BCS system that can be used in farms. An example e.g. that is not covered is how the final BCS score is connected to the ID of the individual cow. Sometimes it is difficult to understand the specific message of sentences. Therefore I advice also to correct it for English. The following comments are restricted to the research method and the interpretation.

Reply: We really grateful for your approval that it is an interesting and important application, and thank you for the pertinent criticisms and insightful suggestions. In the first stage of our work, it really focuses on how to score body condition of each cow. In the future work, we will connected the pedometer data with our cameras. On our cooperative farm of Huahao, each Holstein cow wears an Afimilk pedometer which can identify the cows. When the cow walk down the milking corridor, the ID of pedometer will be recorded, and then the identity association work must be done at this time in the future. We described this future work from Line 466 to 468. (More detailed responses are in the attachment.)

Thank you for your criticisms of English, for not the native speakers, we will try our best to rephrase some sentences and statements. We revised the manuscript according your suggestions. Each modification in the manuscript is highlighted in red.

(2)  L2:  Why not use full text of SSD 'Single Shot Detector'

Reply:  Thank you for your good advice, and I changed the title with the full text of SSD: “An improved single shot multibox detector method applied in body condition score for dairy cows”.

It should be note that “Single Shot multibox Detector” is often abbreviated as “SSD” in most references. Here, the word first appeared in the title, we agree with you very much.

(3)  L19:  do not use the word production. The system monitors Body Condition

Reply:  Thank you, we have changed following your suggestion. Changed to “monitor the BCS of dairy cow remotely”

(4)  L26: are veterinarians the only people scoring in the traditional methods or can it also be done by other people like farmers.

Reply:  Thank you, all the people who have been trained can do this work. We have changed it to “rely on veterinary experts or skilled staffs”.

(5)  L28: change 'wildly' in 'widely'

Reply:  Thank you, it our mistake, we have changed it to “widely”.

(6)  L57: do you have some reference to the 'person to person' differences. This is very important for your research, since you worked with two veterinarians, but no information is given on their differences and quality to score BCS. Just being a veterinarian might not be enough qualification.

Reply:  Thank you very much! This view has indeed been confirmed by several references. It is our negligence that we should cite references. And we have cited the reference [3, 5].

Our data was scored by two veterinarians who were working in Huahao Farm, the BCS result was a synthesis of their two evaluations. Even we did this arrangement, there were still subjective errors. We introduced this phenomenon in Discussion Section from Line 466 to Line 482.

There is no absolutely correct data set in BCS assessment. At present, we can only trust the BCS scored by veterinary experts because human classification is the most intelligent.

(7)  L88: can you elaborate on the 'experimental comparison' since this is the only explanation why you need more algorithms.

Reply:  Thank you very much! We have rephrased it as “For experimental comparison, YOLO-v3 is also introduced because it’s another more recent deep learning method based on regression”
In our study, we introduced an improved SSD method. First of all, it must be compared with the original SSD, to show the effect of improvement. And YOLO-v3 is one more recent deep learning method which is also regression-based method.

These three methods comparison are persuasive to support our work. If we only employ our SSD method, the performance (advantages or disadvantages) are not obvious.

The experiments show that the improved SSD method can achieve 98.46% classification accuracy and 89.63% location accuracy which is a little lower than the original SSD, but it has: (1) faster detection speed with 115 fps; (2) smaller model size with 23.1MB compared to original SSD and YOLO-v3, these are significant advantages for reducing hard ware costs.

(8)  L90: is the statement 'a regular camera is enough' enough or do you have some minimum requirements on your images and that 2D camera with enough fps, resolution and light is in line with your requirements.

Reply:  Sorry, we have rephrase the statements that “and a common camera (with 3 megapixel, 1280*720 resolution, 25 fps, and working in nature light) is enough to capture 2D images of some key body parts of dairy cows.”

In addition, we described our experiment requirements in section 2: Materials and Methods.

(8)  L95-106: This text is not needed in the introduction.

Reply:  Thank you, we have followed your suggestion, and we have removed this paragraph.

(9)  L115-116: The first sentence is a repletion and not needed in this context.

Reply:  Thank you very much, we have followed your suggestion, and we have removed this sentence.

(10)  L123: change (a) and (b) in left and right. You can argues whether you need the (b) part since the labelling is done based on the (a) part.

Reply:  L123 in new manuscript is Line 113. We have changed (a) and (b) in “left” and “right”. And the right half of the figure is important because in Section 2.1, we want to introduce the principle of BCS, the relevant parts of cows can be fully displayed by side view and back view. Although the side view of the cow image that we did not use, the readers can be easier to understand the relationship between BCS and cow’s body. In addition, the paper mentioned “ribs” several times, it can be found in the right half of the figure.

(11)  L128-129: you adopt the 5 point American scale because it only has 5 points! Is this reason enough. Based on this choice I have a more general point. You adopt the 5 point scale, but in your results you also present the intermediate .5 points (see figure 4 etc). This is not in line with each other.

Reply:  L128-129 in new manuscript is Line 119-120. We are sorry, but we want to explain: 5-piont scale system is widely used in the world, we reference to several literature, such as [5, 13, 15, 16, 24, 25]. Although we adopt 5-piont scale, in order to achieve higher BCS accuracy, 0.5 points are needed. Juan et al. [17] did a more detailed partition under 5-point scale, their accuracy reached 0.25 point. We list their data as following: (Please see the picture in the attachment.)

We change our description “In this study, we adopt American 5-point scale system to estimate BCS, because there are only 5 rating levels and it is widely used in the world. According the key areas in Figure 1, 5-point scale can be divided into 5 levels. For example, the outline of spine can be divided into: clear, visible, slight, flat and full. These levels reflect the BCS values from 1 to 5.” to “In this study, we adopt 5-point scale system which is widely used in the world. In order to achieve higher BCS accuracy, the intermediate .5 point is employed.”

(12)  L142-149: how were the practical situations where the camera is used. Was there enough light, a good FOV and how many frames per second were measured. This might also be a good place to say something on the connection between the camera measurement and the ID of the cow.  Is it correct that you have the chance with 1500 cows and three times of milking per day to extract 4500 observations per day. You work with 8972 images and measured a couple of months, which means that most of the images could not be used. Please elaborate on this aspect in material and method or in the discussion.

Reply:  Thank you very much! We added the description “It works in indoor environment with natural light, and it aimed downward to the milking passage. The frame frequency is set to 25 FPS, and the resolution and size are set to 1080×720dpi.” in Line 130-132.

We did not carry out the work of identity recognition as mentioned in reply to comment (1), but this work must be done for our BCS system in the future.

We capture 3 times every two weeks. After data acquisition, we spent few weeks to extract frames, select the images with cow tail, find the cows with the recorded scores, reach consensus for the different values, and the labeled a rectangle on each images manually. This complicated work really spent us a few months.

Finally, veterinary experts labeled the images, and then categorize into 5 folders (BCS from 2.5 to 4.5), this data set can be final used for training for deep learning methods.
So, we followed your suggestion and rephrased some sentences from Line 144 to 146, and Line 154 to 157.

(13)  L168: 'we select 3 distinct images' who is doing this and how this is done. What were the criteria to select the images?

Reply:  L168 in new manuscript is Line 161. The veterinary experts labeled the images, and then categorize into 5 folders (BCS from 2.5 to 4.5), we select 3 images from 3 folders. There are no standard rules, 3 images in Figure 3 only selected from three folders to represent three categories. In Figure 3(a), (b), (c) were marked with “2.5”, “3.5”, and “4.5” beside the green boxes.

(13)  L172: The angle of the camera and the observations seem to deliver pictures from an angle that observers are not used to score BCS. How did the observers cope with this and were they capable of scoring BCS in a proper way.

Reply:  L172 in new manuscript is Line 164. Although the angles are different between the camera and the observers, the image captured by the camera will be labeled by the observers and mark the scores manually. The classification ability of human is robust, e.g., we can recognize somebody’s face by side view or even top view.

Deep Neural Network (Machine learning) can learn this rule if the data set is enough. It has this ability to cope this difference when human labeled this image is “2.5”, “3.0”, or others. Maybe this is the charm of AI.

(14)  L177: Labelling was done manually. This means that the two veterinarians only looked at the images and that there is no real BCS score in the barn itself. You have to mention something on this issue. If you later on talk about specificity etc then it should be clear that it is compared to the observation of human looking at images and not compared to humans doing the BCS measurement in the barn.

Reply: L177 in new manuscript is Line 157. Thank you very much! Your suspicions about “score on images” is reasonable, because the camera cannot capture the whole body of the cow. Actually, during the acquisition of the cow images, the two expert scorer was standing aside and rating the BCS of the cows simultaneously. We need the help of them to preprocessing the data, such as marking the tail position in rectangle, and rank the BCS levels. Thus, machine learning methods can find the rules to learn. The similar work was carried out in reference [14,15,16,17].

We listed reference [17], our topic are similar to theirs. (Please see the picture in the attachment.)

The two veterinary experts are the staffs of Huahao Farm, the rated the cows and if the scores are not the same, they would communicate and reach consensus. We spent a lot of time on data set preprocessing, and finally we received 8972 images that they sent to us.

About the data set and BCS scoring work by experts manually, we also discussed from Line 409 to Line 429.

(15)  L207: Is the top line on 'extra feature layers' too short and should it cover also the blocks Cov6 an Conv7

Reply:  L207 in new manuscript is Line 197. Thank you very much, we have redrew the figure, and it covered the blocks Cov6 and Conv7.

(16)  L223:  the whole paragraph on the improved SSD model is a very technical description of the SSD model and for animal scientist difficult to read and to judge whether it is smart to do it in this way.

Reply:  This section introduced improved SSD algorithm, it really focus on mathematical theory and computer science. Our topic is using machine vision and machine learning method to assess BCS automatically for dairy cows, so the computer technology is unavoidable.

(17)  L297: This paragraph gives too much detail. Sometimes a good reference should be enough. Looking to the results more performance indicators are described in this paragraph. Omit the ones that you are not using in the results.

Reply:  L297 in new manuscript is Line 285. Thank you very much! We have removed “True Positives”, “True Negatives”, “False Positives”, “False Negatives”, “Precision”, “Recall”, and “Average Precision”. Only retained seven necessary performance indicators.

(18)  L335:  why these 4 algorithms and give a good reference to the 3 existing.

Reply:  L335 in new manuscript is L307. Thank you for pointing out the mistakes. We had change the original expression “we carried out four algorithms that are our SSD, the original SSD, YOLO-v3, and mobile-SSD algorithm” to “we carry out three algorithms that are our SSD, the original SSD, and YOLO-v3”

(19)  L359-360:  this seems to be part of Material and Method

Reply:  L359-360 in new manuscript is L331-332. Thank you, we rephrased the expression “Firstly, the original SSD, YOLO-v3 and our SSD are required to verify the position performance. Position is the ability that how accurate the algorithms find the tail of dairy cow. To answer this question, nine image samples are randomly selected from the test dataset to compare the performance of the three algorithms.” as “To see the location performance, nine image samples are randomly selected from the test dataset to compare the performance of the three algorithms”.

(20)  L383:  as with other figures. Give also description of the axis. What are the units? you present.  Why only the results of three algorithms. What happened with the mobile-SSD algorithm.

Reply:  L383 in new manuscript is L352. We have given the description of the axis. For the Y axis is the IoU percent, and X axis is the image index, so there are no units. Mobile-SSD algorithm was not carried out, this mistake was the same as item (18).

(21)  L385: Why use a pie chart. The same information can be presented in the text or in a short table.

Reply:  L385 in new manuscript is L355. Thank you very much! We have followed your suggestion, and changed the pie-chart to a short table.

(22)  L390-399: this is a mix between results, discussion and even conclusion. Stick to present the results in this part of the paper. Do the discussion in chapter 5.

Reply:  L390-399 in new manuscript is L357-363. We rephrased this paragraph following your suggestion. We changed it as follows:

In order to verify this performance of the above models, classification accuracy (CA) is employed. Table 3 shows the CA for each class (or BCS values) in test set. For each BCS level, it can be drawn a conclusion that the extreme BCS levels (referred to BCS 2.5 and BCS 4.5) have a higher accuracy than the middle BCS levels in all the three algorithms. The last row of Table 3 is mAP for each method. The mAP of our SSD is 98.46%, the mAP of original SSD is 99.10%, both of them have a higher accuracy than YOLO-v3 of which the mAP is 88.84%. The mAP of our SSD is a little (0.64%) lower than the original SSD.

(23)  L413:  The table might be misleading since you give absolute numbers. Then I expect that the totals per BCS (score) are 99, 254, 212, 207 and 126. There are different totals between the models. Think of presenting it as percentages and explain well in the text why you have different total.

Reply:  L413 in new manuscript is L380. We are sorry, but we want to explain why the totals are different between models. Because in deep learning algorithms, there are predict bound boxes for each image, and there is a threshold, the threshold was set to 0.6 in our study, for each algorithm, and there were 3 situations:

(a) Threshold≥0.6, there was no predict bound box for this image. This situation caused that the totals per BCS in Table 4 were less than the totals per BCS in Table 1.

(b) Threshold≥0.6, there was one predict bound box for this image.

(c) Threshold≥0.6, there were more than one predict bound boxes for this image. It resulted in that the totals per BCS in Table 4 were more than the totals per BCS in Table 1.

Secondly, confusion matrices are often used in target detection and classification algorithms. In our study, the percentages form was presented in Table 3 which was the results of the classification accuracy (CA) for per BCS.

Thank you very much, for your misunderstanding, we add a new section in discussion from Line 398 to Line 408. (Pictures are in the attachment.)

(24)  L421: I cannot see in the table where the BCS 2.5 is wrongly sorted into 3.5 for the original model. Maybe you are talking about the YOLO-v3 model.

Reply:  L421 in new manuscript is L378. Yes, it is about YOLO-v3, we rephrased this paragraph to eliminate misunderstandings. We changed it as “In YOLO-v3 cell, one image of BCS 2.5 is wrongly sorted into 3.5 category with 1.0 units of error”

(25)  L442: what are the units in the axis.

Reply:  L442 in new manuscript is L396. Thank you very much! We have changed Figure 9 and Figure 10 to Table 5.

(26)  L446-450:  is this summary needed?

Reply:  Thank you, we have removed this summary.

(27)  L452-466:  this looks more a explanation of the results and is hardly a discussion. This seems to be for the whole discussion chapter. Try to bring in some references and e.g. can veterinarians really score on images.

Reply:  Thank you very much! We removed most of the original paragraph, and we added new discussion. We followed your suggestion and added a section for “Can veterinarians score on images?”

(28)  L501: 'manually scoring stage by veterinary experts' say that they score images and not in the barn.

Reply:  L501 in new manuscript is L363. No, it is not score image. It referred the introduction section to express that the history of BCS development: BCS by human and BCS by machine. We have rephrased this paragraph following your next suggestion (29), and this sentence was removed.

(29)  L511-513: this sentence is the conclusion. The rest of this paragraph reads more like a summary. So it can be much shorter.

Reply:  L511-513 in new manuscript is L469-481. Thank you very much! We have rephrased conclusions and reduced the paragraph from 17 lines to 12 lines. And we have cut down some summary statement.

(30)  L533: In the references 7, 12, 13,22, 24, 26, the words et al are used. Why not give all the names of the authors.

Reply:  Thank you very much! We have followed your suggestion and given all the names of the authors which were highlighted in red.

(31)  L561: It seems that a new reference starts at Thi Thi Zin.

Reply:  L561 in new manuscript is L526. Thank you very much! It is our mistake, we have given a new index for this reference. All the references are reordered.

(32)  L566: is it sufficient with website reference (see also 29 and 30) to give the link, or is it also needed to give the extraction dat.

Reply:  L566 in new manuscript is L529. Thank you very much! Reference [12] need to be retained, because it is his official website and we cannot find the description of the DeLava BCS scanner in any other references. Reference [29] and [30] has been removed. We changed Reference [12] as following:

[12] DeLaval Body Condition Scoring BCS. Available online: http://www.delavalcorporate.com. (Accessed on March 25, 2019)

Reviewer 2 Report

The paper is interesting, but the language used in the introduction needs  to be improved. The references are reported in the wrong order. I recommend a language revision before resubmitting the paper.

Abstract:

L28: “wildly”? widely.

Introduction

L49-50: please rephrase.

L55-56: Ribs not rips.

L61: “There exists a certain correlation between cow’s weight and its BCS” this seems to be in opposition to what is written in L45: “It has no direct correlation with body weight and frame size”. Please clarify

L66: What do you mean with “sitting knots?”

L67: maybe is better blocked than fixed.

L68: please use “carried out” instead of “carry out”

L69-83: Please correct the bibliographic order. Indeed, theauthors included the number11 at the number 10 (L561) and subsequently the order is completely wrong. Futhermore, re – order from refence 14 to 17. (bibliographic records have been mixed). Please check all the bibliographic record in introduction.

L100: points instead of ponits.

L118: Probably chute is not the correct word, try to use corridor.

L120. Probably feeling is not the correct word, try to use evaluate.

Materials and Methods

In general: please use the same verbal tense. Don’t mix past with present.

L152: remove “by”

L160-L161: Please rephrase. The meaning is not clear.

L172: Correlation instead of correlationship

L181: were the test set images chosen after the labelling procedure? Please clarify.

L296: is the layout of the table correct for this journal?

Results

L466 please use “overfed” instead of “overfeeding”

References

L546: Condition instead of dondition

L606: “Earle DF. A guide to scoring dairy cow condition [J]. Journal of Agriculture, Victoria, 1976, 74: 228-231” is this reference correct

Author Response

Thank you for your review and for your comments concerning our manuscript entitled “An improved SSD method applied in body condition score for dairy cows”. Manuscript id is animals-527248. Those comments are all valuable and very helpful for revising and improving our paper, as well as the important guiding significance to our researches. We have studied comments carefully and have made correction which we hope meet with approval. The main corrections in the paper and the responds to the reviewer comments were highlighted in red in the new manuscript. We will keep in touch with you and submitt other needed materials.

The main corrections in the paper and the responds to the reviewer comments are as following:

(1)  Comments and Suggestions for Authors

The paper is interesting, but the language used in the introduction needs to be improved. The references are reported in the wrong order. I recommend a language revision before resubmitting the paper.

Reply:  We really grateful for your approval that it is an interesting application, and thank you for the pertinent criticisms and insightful suggestions. According to your comments, we have checked the introduction carefully, and rephrase some sentences and statements, and we are sorry that the references are out of order, then we have corrected them. We revised the manuscript according your suggestions. Each modification in the manuscript is highlight in red.

(2)  L28: “wildly”? widely.

Reply:  Thank you, we have changed it. It is “widely”.

(3)  L49-50: please rephrase.

Reply:  We have changed to “Gillespie et al. [4] found that feed cost accounted for 60% to 65% of the total cost, but more than 10% of the feed was wasted because of overfed.”

(4)  L55-56: Ribs not rips.

Reply:  Thank you very much! We found that the words were all wrong in the text and in Figure 1. We have changed them.

(5)  L61: “There exists a certain correlation between cow’s weight and its BCS” this seems to be in opposition to what is written in L45: “It has no direct correlation with body weight and frame size”. Please clarify

Reply:  Thank you, it was very careful of you to find the problem. We did refer to two literatures. We found they were a bit conflicted, it’s our mistake. After careful consideration, we changed L45: “It has no direct correlation with body weight and frame size” to “, regardless of body weight and frame size”.
Now we list the two paper some related statements highlighted in red box in the attachment.

(6)  L66: What do you mean with “sitting knots?”

Reply:  We found many figure legends, and finally we found this one. It referred the back and rump of cow body. It is our translation error, now we change it to “pin” and “tail setting”. (Please see in the attachment)

(7)  L67: maybe is better blocked than fixed.

Reply: Thank you, we have changed it to “blocked”.

(8)  L68: please use “carried out” instead of “carry out”

Reply: Thank you, we have changed it to “carried out”.

(9)  L69-83: Please correct the bibliographic order. Indeed, theauthors included the number11 at the number 10 (L561) and subsequently the order is completely wrong. Futhermore, re – order from refence 14 to 17. (bibliographic records have been mixed). Please check all the bibliographic record in introduction.

Reply:  We are sorry that the references are misorder, and we have corrected them.

(10)  L100: points instead of ponits.

Reply: Thank you very much for helping me to point out this mistake! Line 95 to Line 106 are deleted following another reviewer’s suggestion.

(11)  L118: Probably chute is not the correct word, try to use corridor.

Reply: L118 in new manuscript is L108. Thank you, we have changed it to “corridor”.

(12)  L120. Probably feeling is not the correct word, try to use evaluate.

Reply: L120 in new manuscript is L109. Thank you, we have changed it to “evaluate”.

(13)  Materials and Methods,  In general: please use the same verbal tense. Don’t mix past with present.

Reply: OK, we have changed to present tense.

(14)  L152: remove “by”

Reply: L152 in new manuscript is L141. Thank you, we have removed it.

(15)  L160-L161: Please rephrase. The meaning is not clear.

Reply:  L160-161 in new manuscript is L147-148. OK, we have rephrased as “The farm has a strict management system, and outsiders cannot enter the core area of milking workshop, such as milking passage in Figure 2”.

(16)  L172: Correlation instead of correlationship

Reply: L172 in new manuscript is L165. Thank you, we have changed it.

(17)  L181: were the test set images chosen after the labelling procedure? Please clarify.

Reply: L181 in new manuscript is L170. Yes, all the image samples are labeled. The labeling work for test set images is for comparison. The three algorithms will predict the values of BCS, and these results need to be compared with the manual method by veterinary experts.

(18)  L296: is the layout of the table correct for this journal?

Reply: L296 in new manuscript is L283. Thank you! Because the content of our table is very special and complicated, and now we change it to fit the format of this journal.

(19)  Results, L466 please use “overfed” instead of “overfeeding”

Reply: Thank you very much for helping me to point out this mistake!  Line 466 in discussion are removed following another reviewer’s suggestion.

(20)  References.  L546: Condition instead of dondition

Reply: L546 in new manuscript is L509. Thank you, we have changed it to “condition”.

(21)  L606: “Earle DF. A guide to scoring dairy cow condition [J]. Journal of Agriculture, Victoria, 1976, 74: 228-231” is this reference correct

Reply:  L606 in new manuscript is L509. Yes, the reference is correct. We read the following paper, in this paper, it cited this reference that “In 1976, Earle et al. firstly proposed the 8-ponit BCS system in Austria”. But we cannot find this paper in many paper database or Google Scholar, maybe the paper published too long ago. But the 8-point scale was first proposed by Earle in Australia.

Reference: Roche, J. R., P. G. Dillon, C. R. Stockdale, L. H. Baumgard, and M. J. VanBaale. Relationships among international body condition scoring systems [J]. Journal of Dairy Science, 2004, 87:3076–3079.

Round 2

Reviewer 1 Report

The paper improved quite a lot and the logic of the story becomes clear. I think that a good check on English by a native speaker will improve readability even more.

L148:  I think the word 'workshop' is not what you intend to say.

L353:  maybe it is better to omit the word 'index' and keep Image. You present the results of the 50 independent images. You can even discuss whether you can use lines for it. maybe a table might also fit and present the original and yolo as deviations from our SSD model.  

L405-407:   You mention three situations. However they are all three initiated if threshold is ≥ 0.6. This seems to be strange. Maybe I expect one initiated by 〉, one by 〈 and one by =

L466: now you focus in general on dairy herds. Maybe this might be too optimistic. At least you have experience with BCS measurements for Holstein cows.

Author Response

Dear Reviewer:

  Thank you for your first review, it help us greatly to eliminate mistakes, and help to improve our writing, we really appreciate your kind help. We followed your last comments and suggestions and resubmitted the manuscript which we hope meet with approval.
